# Diagnostic Value of Salivary Amino Acid Levels in Cancer

**DOI:** 10.3390/metabo13080950

**Published:** 2023-08-15

**Authors:** Lyudmila V. Bel’skaya, Elena A. Sarf, Alexandra I. Loginova

**Affiliations:** 1Biochemistry Research Laboratory, Omsk State Pedagogical University, 14 Tukhachevsky Str., 644043 Omsk, Russia; sarf_ea@omgpu.ru; 2Clinical Oncology Dispensary, 9/1 Zavertyayeva Str., 644013 Omsk, Russia; pai198585@mail.ru; 3Department of Oncology, Omsk State Medical University, 12 Lenina Str., 644099 Omsk, Russia

**Keywords:** amino acids, metabolome, saliva, cancer, diagnosis

## Abstract

This review analyzed 21 scientific papers on the determination of amino acids in various types of cancer in saliva. Most of the studies are on oral cancer (8/21), breast cancer (4/21), gastric cancer (3/21), lung cancer (2/21), glioblastoma (2/21) and one study on colorectal, pancreatic, thyroid and liver cancer. The amino acids alanine, valine, phenylalanine, leucine and isoleucine play a leading role in the diagnosis of cancer via the saliva. In an independent version, amino acids are rarely used; the authors combine either amino acids with each other or with other metabolites, which makes it possible to obtain high values of sensitivity and specificity. Nevertheless, a logical and complete substantiation of the changes in saliva occurring in cancer, including changes in salivary amino acid levels, has not yet been formed, which makes it important to continue research in this direction.

## 1. Introduction

Amino acids, as a raw material for protein synthesis and a product of protein metabolism, enter the body or are synthesized endogenously. They play mainly physiological roles as major metabolites and regulators of metabolism among the most important compounds.

Amino acid metabolism is part of the altered processes in cancer cells [1]. Amino acids are the building blocks as well as sensors of signaling pathways that regulate basic biological processes. The main role of amino acids is to provide substrates for the biosynthesis of proteins and nucleic acids and to participate in the metabolism of carbohydrates and lipids. They are also involved in non-enzymatic antioxidant mechanisms (through the synthesis of glutathione) and epigenetic modifications (mainly with the participation of S-adenosylmethionine as a methyl group donor) [2,3,4]. Pathways of amino acid metabolism that tumor cells activate as antioxidants have been described, including the metabolism of cysteine and methionine and their association with folic acid, transsulfuration pathways, and ferroptosis [5]. Amino acids are actively studied as potential targets for anticancer therapy (asparagine, arginine, methionine and cysteine) [1,6,7]. Asparagine depletion has been successfully used for decades in the treatment of acute lymphoblastic leukemia; arginine auxotrophic tumors are excellent candidates for treatment with arginine starvation, etc.

A number of studies have shown that amino acids used as potential biomarkers vary for different types of cancer, and changes in the concentration of amino acids in body fluids and tissues are important for diagnosing cancer, as well as choosing treatment tactics [8,9,10].

One of the promising biological fluids for the determination of amino acids is saliva [11,12,13,14,15,16,17,18,19]. It contains molecules that can potentially be associated with the course of the disease and facilitate diagnosis and prognosis, including proteins, mRNA, miRNA, enzymes, hormones, antibodies, antimicrobial components, growth factors, and metabolites [20]. The determination of amino acids in saliva has been described in the norm [21,22], including intra-day and inter-day variations, as well as the age of the volunteers [23]. The existing literature data on the content of amino acids in saliva in cancer patients are few, scattered and contradictory. There is currently no detailed justification for changes in the concentrations of individual amino acids and/or their combinations in saliva in certain types of cancer, including no understanding of how these changes are related to the content of amino acids in cancer cells, which emphasizes the relevance of research in this direction. In this review, we summarized the results of studies on the determination of the amino acid composition of saliva in various types of cancer.

## 2. Materials and Methods

A review was conducted up to 15 May 2023 using the databases PubMed, Scopus and Web of Science. The search sentence used was TITLE-ABS-KEY (cancer OR carcinoma OR neoplasm OR tumour OR tumor OR oncology) AND saliva AND (amino acids OR metabolomics). Records were screened by title, abstract, and full text by two independent investigators. At the initial stage, duplicate articles in different databases were deleted; reviews and conference abstracts were excluded from consideration.

The search returned a total of 156 reports from databases after deleting duplications. We then excluded studies that were not related to the study question (the matrix did not fit the query; no saliva; the participants were not human) or were reviews, conference papers, book chapters, notes, letters or editorials. The final list contained 21 papers.

## 3. Results

Basic information about the studies included in the review is given in Table 1.

Saliva collection techniques were similar across all studies. Saliva samples were collected without additional stimulation in the morning (7:00–8:00 [26], 8:30–10:30 [34,35,40,44], 9:00–10:00 [25], 9:00–11:00 [36,46], strictly at 8:00 [30]). Subjects were asked to refrain from eating, drinking, smoking, or using oral hygiene products for at least 1 h prior to saliva collection; in some cases, patients did not eat for 12 h. Saliva samples were collected 5–10 min after rinsing the mouth with water by spitting into a plastic container in an amount of 100 μL to 5 mL, which took an average of 5–15 min. In one study, DNA Genotek OMNIgene ORAL OM-505 (Ottawa, Ontario) was used to collect saliva [45]; in another, Salivette^®^ polyester swabs (SARSTEDT, Germany) were used by the 5 min soak method [44]. Immediately after collection, saliva samples were centrifuged at low speed (1500–3000 rpm) for 10–20 min [24,25,26,27,28,29,30,31,44] or high speed (12,000–13,500 rpm in for 20–30 min [34,35,38] at 4 °C to remove any insoluble materials, cellular debris, and food debris. In one study, additional filtration of contaminants in saliva was performed using a centrifugal filter (Nanosep^®^; Pall Corporation, Port Washington, NY, USA) [30]. In studies [32,33,36,37,42,43,46], saliva was frozen without prior centrifugation. In studies [26,40,41], proteins were pre-sedimented before freezing. If it was impossible to perform a saliva study on the day of collection, the samples were stored at −35–40 °C [27,28,29,34,35,44], −70 °C [38], and in other cases at −80 °C. Further sample preparation was determined by the method of analysis.

The amino acid composition of saliva is determined independently via liquid chromatography [26], as well as in the course of complex analysis of metabolites using gas chromatography coupled to a mass spectrometer (GC-MS) [33,45], nuclear magnetic resonance (NMR) spectroscopy [42], and various variants of mass spectrometry [24,25,27,28,29,30,31,32,34,35,36,40,41,43,44,46]. Also surface-enhanced Raman scattering (SERS) Sensors using metal nanoparticles, especially gold nanoparticles (AuNPs) and silver nanoparticles [38], DNA/Ag NCs-based biosensing system [37], carbon dots confined in N-doped carbon (CDs@NC) for colorimetric detection of D-Pro and D-Ala [39] is used; however, in these variants, one or a few amino acids are determined.

The main part of the studies is devoted to the diagnosis of oral cavity cancer (8/21), breast cancer (4/21), gastric cancer (3/21), lung cancer (2/21), glioblastoma (2/21) and one study concerns the of diagnosis colorectal cancer, pancreatic, thyroid and liver cancer, respectively (Table 1).

It is shown that more than 50% of studies mention the amino acids alanine and leucine, while none of them pays attention to asparagine and cysteine (Table 2). Combinations of amino acids in the studies of different authors vary; however, in addition to alanine and leucine, an important role is noted for valine, isoleucine, phenylalanine, and proline.

The content of amino acids varies differently depending on the type of cancer; for example, for lung and thyroid cancer, the level of amino acids is lower than for healthy controls and higher for breast cancer (Table 2). For oral cancer, the data vary, which may be due to the characteristics of the sample or the method of sample preparation and analysis since the list of methods is quite wide (Table 1).

Only three studies show the absolute concentrations of amino acids in saliva in the norm [26,27,34], while the data are given in different units of measurement (Ref. [26]—µmol/mL, Refs. [27,34]—ng/mL). It was also noted that the range of variation in amino acid concentrations in the norm is quite wide; for example, in [34], the concentration of alanine was 11,424.5 ± 12,290.0 ng/mL, leucine—1072.3 ± 1344.7 ng/mL, etc. Such wide ranges of variation require the collection and analysis of more representative samples, as well as the inclusion of a healthy control group in the design of each experiment, in order to exclude the influence of methodological inaccuracies on the result of the analysis. More often, the authors give the relative values of the content of individual amino acids, determined in comparison with the control group (Table 3). According to the given values, it is possible to compare the results of different authors on the content of amino acids for the same type of cancer (oral cancer and breast cancer, Table 3). It is shown that the data for breast cancer are more similar to each other than for OSCC (Table 3). All other scientific papers present the results of data processing by multivariate statistics without analyzing the original data, so it is not possible to assess their reproducibility.

Table 4 shows the values of sensitivity and specificity when using individual amino acids for the diagnosis of different types of cancer. Sensitivity and specificity values of 75.1 and 71.8, respectively, were obtained (Table 4), while when using combinations of amino acids with each other or with other metabolites, sensitivity and specificity values increase (Table 5).

The main methods of multivariate statistics used to analyze the amino acid profile of saliva in cancer are principal component analysis (PCA), partial least squares-discriminant analysis (PLS-DA), orthogonal partial least squares-discriminant analysis (OPLS-DA), hierarchical and heatmap cluster analysis, logistic regression (LR) (Multiple—MLR, binary—BLR), ANN and Random Forest models, classification and regression tree method (CART) (Table 5).

In 13 studies out of 21 (Table 5), combinations of metabolites and amino acids are used to build a diagnostic algorithm. At the same time, only four algorithms use amino acids directly, while all the others use a wider set of parameters for construction. Most often, the algorithms include valine, alanine, leucine and phenylalanine; a little less often, serine, tyrosine, histidine, lysine, glutamic acid, glutamine, threonine, isoleucine and proline (Table 4). It should be noted that it was for valine and alanine that higher AUC-ROC values were obtained (Table 5).

Several amino acids have been proposed for early cancer detection: leucine for OSCC [27,28,29]; proline, threonine and histidine were included in the complex index for detecting early breast cancer [34]; serine, proline, valine and arginine in combination with other metabolites for lung cancer [40]. Only one study reported absolute concentrations of amino acids in the early and advanced stages of breast cancer (Table 3) [34].

## 4. Discussion

The researchers found that amino acid metabolism was markedly altered in cancer cells [47] and plasma free amino acids also had a visible difference between cancer patients and healthy controls [48]. So, leucine, isoleucine and valine are three essential branched-chain amino acids that are raw materials for the synthesis of proteins and nucleic acids with the formation of acetyl and/or propionyl coenzyme-A, which enter the tricarboxylic acid cycle in the last stages of their metabolic pathways. Proline is a regulator of cytoplasmic balance and an important component of collagen, the most abundant protein in the body. Its biosynthetics are required for remodeling of the tumor microenvironment and extracellular matrix to promote reprogramming and proliferation of cancer cells. In addition, proline can generate adenosine triphosphate for cell growth through the tricarboxylic acid cycle during catabolism. Glycine is a precursor for the synthesis of important substances such as proteins, nucleic acids and lipids that are essential for cell growth. Meanwhile, their biosynthesis will affect the antioxidant capacity of cells to maintain tumor homeostasis. Studies have shown that mitochondrial glycine metabolism is closely related to the rapid proliferation of cancer cells [49]. Alanine is an amino acid in glycogen; it is the main source of carbon rather than the carbon of glucose in the tricarboxylic acid cycle (production of lipids and non-essential amino acids), which makes glucose more available for the biosynthesis of other functions; this synergistic metabolism promotes tumor growth. Phenylalanine is converted to tyrosine via catalytic oxidation by phenylalanine hydroxylase, and tyrosine is involved in the metabolism of glucose and fats in the body, which are the main energy sources for rapidly growing cancer cells. Methionine is a substrate for the biosynthesis of the universal methyl donor S-adenosylmethionine, and threonine maintains the intracellular concentration of S-adenosylmethionine and methylation of histones that enter the folate cycle by converting to folate intermediates. The folic acid cycle and the methionine metabolism cycle are the two main components of one-carbon metabolism important for the growth and multiplication of cancer cells at a high growth rate. Tryptophan, a precursor of kynurenine, is metabolized to kynurenine by indolamine 2,3-dioxygenase. Under normal physiological conditions, indolamine 2,3-dioxygenase expression is modulated; however, it is frequently activated in some types of cancer [50]. Upregulation of indolamine 2,3-dioxygenase expression leads to increased tryptophan metabolism, which increases kynurenine production. Kynurenine, an oncometabolite, inhibits T-cell differentiation and therefore promotes cancer growth and development. This ultimately leads to a decrease in serum tryptophan levels in cancer patients. Several studies have suggested dependence of tumor cells on glutamine [51,52].

Thus, changes in the content of amino acids in biological fluids, including saliva, in oncological diseases are metabolically substantiated and deserve the attention of researchers. It should be expected that in addition to the general patterns for each individual type of cancer, characteristic combinations of amino acids could be identified.

**OSCC.** Sugimoto M. et al. [24] identified the following amino acids for the diagnosis of oral cancer: alanine, leucine, isoleucine, glutamic acid, phenylalanine, and serine. Wei J. et al. [25] focused on phenylalanine and valine, which had also previously been identified as discriminating serum metabolites in OSCC compared to healthy subjects [53]. At the same time, amino acid concentrations decreased in cancer, which may be associated with increased glycolysis during cell proliferation in cancer tissues [24]. Reddy I. et al. [26] showed that levels of amino acids histidine, threonine, valine, isoleucine, methionine, phenylalanine, leucine, lysine, tyrosine, arginine, alanine, glycine, serine and aspartic acid were significantly higher in both well-differentiated OSCC cases and moderately differentiated OSCC cases than in healthy controls. Wang Q et al. [27,28,29] showed that the content of phenylalanine and leucine decreased compared to the control. Mean concentrations of phenylalanine in OSCC patients with T1–2 compared with healthy controls were 1.6 times lower (*p* = 0.028) and leucine 3.8 times lower (*p* = 0.001). As a standalone biomarker, leucine may have a better predictive power for the early stages of OSCC, and phenylalanine can be used to screen for and diagnose advanced stages of OSCC. The combination of phenylalanine and leucine improved sensitivity and specificity. Ohshima M. et al. [30] isolated valine, leucine, isoleucine, tryptophan, and alanine, while Lohavanichbutr P. et al. [31] focused on glycine and proline. The concentrations of glycine and proline in the saliva of oral cancer patients are lower than those of the control group in both sets of samples. The authors hypothesized that OSCC tumor cells take up glycine from the salivary extracellular space and actively synthesize glycine in mitochondria to form one-carbon units for subsequent nucleotide synthesis to support tumor progression. Yatsuoka W. et al. [32] found significant differences between the groups in terms of histidine and tyrosine levels, and de Sá Alves M. et al. [33] isolated methionine and leucine in a set of 25 metabolites. Thus, most studies point to the important role of leucine and isoleucine, phenylalanine, valine and alanine in the diagnosis of oral cancer. At the same time, in different studies, the concentration of amino acids varied in different ways; no definite regularity had been revealed: in half of the studies, the concentration of amino acids was up-regulated, and in the other half—down-regulated. Apparently, this was due to the characteristics of the studied sample and/or differences in sample preparation and analysis of the biomaterial. Three of the four studies for which amino acid concentrations were up-regulated used capillary electrophoresis time-of-flight mass spectrometry (CE-TOF-MS), while ultra-performance liquid chromatography was used to obtain down-regulated amino acid profiles. However, a more detailed consideration of the process of sample preparation and analysis was beyond the scope of this review.

**Breast cancer.** Sugimoto M. et al. [24] proposed eight amino acids (lysine, threonine, leucine + isoleucine, glutamic acid, tyrosine, valine, and glycine) as part of a number of metabolites for diagnosing breast cancer. Cheng F. et al. [34] analyzed 17 amino acids to distinguish stage I–II breast cancer from healthy controls: Pro, Thr, and Glu (*p* < 0.001); Phe, Trp, Met, Asp, Ser, Gln and Leu (*p* < 0.01); His, Val and Lys (*p* < 0.05); and Ala and Arg (*p* > 0.05). However, only three amino acids were included in the complex index for detecting early breast cancer: proline, threonine and histidine (Table 4). Comparison of amino acid levels in stage I–II and III–IV breast cancer showed no significant differences, with the exception of valine (*p* = 0.027). Zhong L. et al. [35] identified two amino acids: phenylalanine and histidine. An obvious decrease in the level of acetylphenylalanine indicates a violation of the metabolism of phenylalanine in individuals with breast cancer. A similar abnormality in phenylalanine metabolism was found in oral squamous cell carcinoma [29]. Murata T. et al. [36] identified four amino acids leucine, glutamine, isoleucine, and serine. The authors also determined salivary metabolite concentration in four cancer subtypes: Luminal A-like, luminal B-like, HER2-positive, and triple-negative. However, differences between molecular biological subtypes of breast cancer are characterized by other metabolites, not amino acids. Thus, in the four studies analyzed, the authors identified different amino acids; in three studies, leucine and isoleucine are common, but there is no justification for the choice of these particular amino acids from the point of view of the biochemistry of the ongoing processes, which has yet to be done.

**Gastric cancer.** For gastric cancer, all studies used spectroscopy methods: Raman and ultraviolet, as well as sensors. Chen et al. did not evaluate the content of individual amino acids (Gly, Gln, His, Ala, Glu, Pro, Tyr); data processing was carried out using principal component analysis; sensitivity and specificity were noted above 80 and 87%, respectively [38]. In [37], the DAA index was proposed, which ranged from 2 to 10 for cancer patients; while in the healthy control group, DAA scores remained low (from 0 to 1). The authors quantified the probe fluorescence shift by defining the percent fluorescence quenching (PFQ) as (I_0_ − I)/I_0_ ∗ 100%, where I_0_—the fluorescence intensity of the blank control, I—the fluorescence intensity of the test samples. The DAA index was evaluated as a number of PFQ/5%. The test results showed that the concentration of DAA in healthy people and stomach cancer patients was 0~25.3 μM, in particular (D-Ala)/0~11.3 μM (D-Pro) and 50.6~253.2 μM (D-Ala)/22.5~112.6 μM (D-Pro), respectively. These results showed that gastric cancer patients and healthy people can be distinguished using the silver DNA nanocluster. Moreover, our test values were in good agreement with the range of physiologically significant concentrations (6.6 ± 1.2 μM (D-Ala)/12.8 ± 5.5 μM (D-Pro) for healthy individuals and 205.8 ± 79.5 μM (D-Ala))/80.3 ± 34.2 μM (D-Pro) for patients with gastric cancer). Li Z. et al. also used sensors; however, unlike [39], carbon dots confined in N-doped carbon (CDs@NC) were used, but the amino acids determined were the same D-Ala and D-Pro. The sensitivity and specificity of these methods have not been evaluated, and no detailed patient information is available. Of note, all three studies highlight alanine and proline as important amino acids for diagnosing gastric cancer.

**Lung cancer.** Jiang X. et al. [40] identified 23 metabolites, which allow diagnosing early lung cancer with high accuracy; this list includes four amino acids (serine, proline, valine, arginine) (Table 4). A decrease in amino acid content and activation of downstream metabolites of amino acid metabolism, including ketoleucine, N-acetylhistidine, imidazolepropionic acid, N-acetylproline, allisin, gentisic acid, 3-hydroxyanthranilic acid, γ-aminobutyric acid and pyroglutamic acid were observed in the early lung cancer group, which is consistent with previous studies [54,55]. This may be due to protein deficiency and increased amino acid requirements caused by tumor growth [56]. According to Takamori S. et al. [41], profiles of 10 salivary metabolites differed markedly between lung cancer and benign lung lesion patients. Among them, salivary tryptophan concentration was significantly lower in patients with lung cancer. It is known that the level of tryptophan in the blood serum in patients with lung cancer was significantly lower than in healthy people [57]. Takamori S. et al. [41] developed an MLR model in which four metabolites were selected, including lysine and tyrosine. The MLR model had a high ability to distinguish between lung cancer and benign lung lesion patients (AUC = 0.729, Table 4). It should be noted that in lung cancer, in both studies, the content of amino acids was downregulated.

**Glioblastoma.** Garcia-Villaescusa A. et al. [42] showed that four amino acids leucine, isoleucine, alanine, and valine change their content in glioblastoma. It is known that alanine is elevated in malignant brain tumors and, therefore, can be used to distinguish between tumor type and grade [58]. In addition, an increase in leucine-rich proteins was observed in some malignant gliomas, which was mainly associated with an increased risk of developing astrocytomas and glioblastomas [59]. A study by Bark J.M. et al. [43] identified valine among a large number of metabolites, which overlaps with the results of García-Villaescusa A. et al. [42] and other studies showing the contribution of valine to cancer diagnosis.

**Thyroid cancer.** We found the only study of amino acids in saliva in thyroid cancer [44]. The authors showed that the content of all studied amino acids in thyroid cancer is reduced compared to healthy controls. The results are consistent with previous blood metabolism studies for the detection of thyroid cancer. Abooshahab et al. described a marked decrease in the concentrations of valine, phenylalanine, proline, glycine, methionine and threonine in patients with thyroid cancer [60]. Huang et al. reported in a comparative study between healthy controls and thyroid cancer that the expression of alanine, proline and tryptophan was reduced in the patient group [61]. Threshold values have been established that can be used for diagnosis (Table 5). The values of sensitivity and specificity for individual amino acids vary within a fairly wide range, the average sensitivity was 76.9% (50.8–100.0%), and the average specificity was 55.1% (43.1–92.2%). When using the complex index, the values of sensitivity and specificity increased significantly and amounted to 91.2 and 85.2%, respectively.

**Colorectal Cancer.** In the only study by Kuwabara H. et al. [46] showed that several amino acids, such as isoleucine, valine, lysine, and alanine, were elevated in both adenomas and colorectal cancer, but the authors do not offer justification for what this may be due to.

**Hepatocellular cancer.** Hershberger C.E. et al. [45] proposed four variants of algorithms for detecting hepatocellular cancer based on metabolomic profiling of saliva, but only one of the algorithms included the amino acids glutamine and serine (Table 4). It has previously been reported that serum serine levels are altered in patients with cirrhosis compared to healthy individuals and in the urine of patients with hepatocellular cancer compared to healthy individuals [62,63]. Glutamine levels differed in serum and liver tissue between healthy people and people with cirrhosis, healthy people and people with hepatocellular cancer [64,65]. The enzyme responsible for glutamine production, glutamine synthetase, has been identified as a potential biomarker for early hepatocellular cancer in proteomic assays and has been shown to promote cell migration by mediating epithelial–mesenchymal transition [66].

**Pancreas cancer.** Sugimoto M. et al. [24] identified two amino acids in five metabolites for diagnosing pancreatic cancer: phenylalanine and tryptophan. Although pancreatic cancer samples showed a trend towards decreasing levels of amino acids, including leucine, isoleucine, valine, and alanine [67], an increase in amino acid levels was observed in saliva. In this regard, the authors argue that further validation of these results by comparing saliva profiles with blood and tissue profiles is necessary in order to understand the reason for the different amino acid profiles in saliva.

## 5. Conclusions

The amino acids alanine, valine, phenylalanine, leucine and isoleucine play a leading role in the diagnosis of cancer by saliva. Depending on the type of cancer, the significance of individual amino acids varies: leucine, isoleucine, phenylalanine, valine and alanine are important for oral cancer, leucine and isoleucine for breast cancer, alanine and proline for gastric cancer, and valine for glioblastoma. Whereas for lung cancer, the list of significant amino acids according to different authors is completely different; however, in both cases, amino acids are down-regulated. In general, amino acid levels in cancer are elevated in all types of cancer except lung and thyroid cancer. The data of different authors on oral cancer are contradictory: the same amino acids in some studies are up-regulated, in others, down-regulated, requiring a more detailed acquaintance with the method of sample preparation and analysis.

In an independent version, amino acids are rarely used; the authors combine either amino acids with each other or with other metabolites, which makes it possible to obtain high values of sensitivity and specificity. Nevertheless, a logical and complete substantiation of the changes in saliva occurring in cancer, including changes in the salivary amino acid level, has not yet been formed, which makes it important to continue research in this direction.

## Figures and Tables

**Table 1 metabolites-13-00950-t001:** Main studies of the amino acid composition of saliva in various types of cancer.

№	Type of Cancer	Author	Method of Analysis	Study Group	Amino Acids (AAs) *
**1**	OSCC	Sugimoto M. et al., 2010 [24]	CE-TOF-MS	OSCC—69, HC—87	Ala, Leu + Ile, Tyr, Glu, Phe, Ser, His, Pro, Lys, Gly, Asp, Gln, Val, Trp, Thr
**2**	OSCC	Wei J. et al., 2011 [25]	UPLC-QTOF-MS	OSCC—37, leukoplakia (OLK)—32, HC—34	Val, Phe
**3**	OSCC	Reddy I. et al., 2012 [26]	HPLC	OSCC—16 (well-differentiated—8, moderately differentiated—8), HC—8	Asp, Glu, Ser, His, Gly, Thr, Ala, Arg, Tyr, Val, Met, Phe, Ile, Leu, Lys
**4**	OSCC	Wang Q. et al., 2014 [27,28,29]	UPLC–ESI–MS	OSCC—60, HC—30	Phe, Leu
**5**	OSCC	Ohshima M. et al., 2017 [30]	CE-TOF–MS	OSCC—22, HC—21	Val, Leu, Ile, Trp, Ala
**6**	OSCC	Lohavanichbutr P. et al., 2018 [31]	HILIC–UPLC–MS	OSCC—101, OPC—58, HC—35	Gly, Pro
**7**	OSCC	Yatsuoka W. et al., 2021 [32]	CE-TOF-MS	Head and neck cancer—9 (underwent radiation therapy)	His, Tyr, Gly, Glu, Asp, Trp, Lys, Met
**8**	OSCC	de Sá Alves M. et al., 2021 [33]	GC-MS	OSCC—27, HC—41	Met, Leu
**9**	Breast cancer	Sugimoto M. et al., 2010 [24]	CE-TOF-MS	Breast cancer—30, HC—87	Ala, Leu + Ile, Tyr, Glu, Phe, Ser, His, Pro, Lys, Gly, Asp, Gln, Val, Trp, Thr
**10**	Breast cancer	Cheng F. et al., 2015 [34]	HILIC–UPLC–MS	Breast cancer—27 (Stage I—5, II—12, III—10)	Leu, Phe, Trp, Met, Val, Pro, Ala, Thr, Glu, Gln, Ser, Asp, Arg, Lys, His
**11**	Breast cancer	Zhong L. et al., 2016 [35]	RPLC-ESI-MS HILIC-ESI-MS	Breast cancer—30 (Stage I—7, II—14, III—8, IV—1), HC—25	Phe, His
**12**	Breast cancer	Murata T. et al., 2019 [36]	CE-TOF–MS	Invasive breast carcinoma—101, Ductal carcinoma in situ—23, HC—42	Leu, Gln, Ile, Ser
**13**	Gastric cancer	Zhang Z. et al., 2017 [37]	DNA/Ag NCs based biosensing system	-	DAA index (D-Ala, D-Pro)
**14**	Gastric cancer	Chen Y. et al., 2018 [38]	SERS sensors	Gastric Cancer (earlier—20, advanced—84), HC—116	Gly, Gln, His, Ala, Glu, Pro, Tyr
**15**	Gastric cancer	Li Z. et al., 2022 [39]	UV–vis absorption spectra	Gastric cancer—5, HC—5	D-Pro and D-Ala
**16**	Lung Cancer	Jiang X. et al., 2021 [40]	MALDI-TOF-MS	Lung cancer—100 (early—89 and advanced—11), HC—50	Ser, Pro, Val, Arg
**17**	Lung Cancer	Takamori S. et al., 2022 [41]	CE-TOF-MS	Lung Cancer—41, benign lung lesion (BLL)—21	Ile, Leu, Lys, Phe, Tyr, Trp
**18**	Glioblastoma	García-Villaescusa A. et al., 2018 [42]	NMR spectroscopy	Glioblastoma—10, HC—120	Leu, Val, Ile, Ala
**19**	Glioblastoma	Bark J.M. et al., 2023 [43]	UPLC-QTOF-MS	Glioblastoma—21	dl-Val
**20**	Pancreatic cancer	Sugimoto M. et al., 2010 [24]	CE-TOF-MS	Pancreatic cancer—18, HC—87	Ala, Leu + Ile, Tyr, Glu, Phe, Ser, His, Pro, Lys, Gly, Asp, Gln, Val, Trp, Thr
**21**	Thyroid cancer	Zhang J. et al., 2021 [44]	HILIC–UPLC–MS	Papillary thyroid carcinoma—61, HC—61	Gly, Ala, Pro, Val, Thr, Leu, Ile, Met, Phe, Trp
**22**	Hepatocellular carcinoma	Hershberger C.E. et al., 2021 [45]	GC-TOF-MS	Hepatocellular carcinoma—37, cirrhosis—30, HC—43	Gln, Ser
**23**	Colorectal cancer	Kuwabara H. et al., 2022 [46]	CE-TOF-MS	Colorectal cancer (CRC)—235, adenoma (AD)—50, HC—2317	Ile, Val, Lys, Ala

Note. *—A complete list of the amino acids that were determined in each study is provided. OSCC—oral squamous cell carcinoma; OPC—oropharyngeal squamous cell carcinoma; HC—healthy control. HPLC—high-performance liquid chromatography, CE-TOF-MS—capillary electrophoresis time-of-flight mass spectrometry, MALDI-TOF-MS—matrix-assisted laser desorption/ionization time-of-flight mass spectrometry, UPLC-ESI-MS—ultra-performance liquid chromatography—electrospray ionization-mass spectrometry, UPLC-MS method in the hydrophilic interaction chromatography (HILIC) and reversed-phase liquid chromatography (RPLC) separations in both positive (ESI+) and negative (ESI-) ionization modes, UPLC-QTOF-MS—ultra-performance liquid chromatography coupled with quadrupole/time-of-flight mass spectrometry.

**Table 2 metabolites-13-00950-t002:** Main studies of the amino acid composition of saliva in various types of cancer.

AA	OSCC								BC				GC			LC		GBM		PC	TC	HCC	CRC	∑
	24 *	25	26	27	30	31	32	33	24	34	35	36	37	38	39	40	41	42	43	24	44	45	46	
**Ala**	↑	↓	↑		↑	↓			↑	↑			↑	↑	↑			↑		↑	↓		↑	14
**Arg**			↑							↑						↓								3
**Asn**																								0
**Asp**	↑		↑			↓	↑		↑	↑										↑				7
**Cys**																								0
**Gln**	↑					↓			↑	↑		↑		↑						↑		↑		8
**Glu**	↑		↑			↓	↑		↑					↑						↑				7
**Gly**	↑		↑			↓	↑		↑					↑						↑	↓			8
**His**	↑		↑			↓	↑		↑	↑	↑			↑						↑				9
**Ile**	↑	↓	↑		↑	↓			↑			↑					↓	↑		↑	↓		↑	12
**Leu**	↑	↓	↑	↓	↑	↓		↓	↑	↑		↑					↓	↑		↑	↓			14
**Lys**	↑		↑			↓	↑		↑	↑							↓			↑			↑	9
**Met**			↑				↑	↑		↑											↓			5
**Phe**	↑	↓	↑	↓		↓	↑		↑	↑	↑						↓			↑	↓			12
**Pro**	↑	↓				↓			↑	↑			↑	↑	↑	↓				↑	↓			11
**Ser**	↑		↑			↓			↑	↑		↑				↓				↑		↑		9
**Thr**	↑	↓	↑			↓			↑	↑										↑	↓			8
**Trp**	↑				↑		↑		↑	↑							↓			↑	↓			8
**Tyr**	↑		↑			↓	↑		↑					↑			↓			↑				8
**Val**	↑	↓	↑		↑	↓			↑	↑						↓		↑	↑	↑	↓		↑	13

Note. *—Study number in the list of references. ∑—number of studies in which this amino acid is mentioned. ↑—amino acid is upregulated; ↓—amino acid is downregulated. BC—breast cancer; GC—gastric cancer; LC—lung cancer; GBM—glioblastoma; PC—pancreatic cancer; TC—thyroid cancer; HCC—hepatocellular carcinoma; CRC—colorectal cancer.

**Table 3 metabolites-13-00950-t003:** Relative change in the concentration of individual AAs in saliva compared to healthy controls.

AA	Oral Cancer (OSCC)				Breast Cancer			Pancreatic Cancer
	[24]	[25]	[26] *	[30]	[24]	[34] *	[36] ****	[24]
**Ala**	3.91		1.85/5.91 **	1.3	1.94	1.68/1.99 ***	~1.5	3.67
**Arg**			4.68/12.6			1.29/1.26	~1.6	
**Asn**								
**Asp**	1.63		6.89/17.2		1.70	2.12/2.09		4.10
**Cys**								
**Gln**	2.35				1.59	2.24/2.55	~2.5	4.96
**Glu**	2.87		0.76/2.01		2.12			4.80
**Gly**	1.38		4.43/8.49		2.32			3.10
**His**	1.70		1.33/2.34		1.35	1.35/1.24		2.02
**Ile**	4.65		7.15/13.4	2.7	3.05		~2.0	7.71
**Leu**			16.1/33.4	2.5		1.81/2.10	~2.5	
**Lys**	1.84		1.63/0.56		2.96	1.90/1.97		3.97
**Met**			13.5/104.4			4.93/2.17		
**Phe**	2.25	0.74	9.54/33.5		1.78	1.67/1.45		3.54
**Pro**	1.63				2.48	3.25/3.97		3.99
**Ser**	1.74		3.74/10.3		1.66	2.62/2.96	~2.2	4.34
**Thr**	2.15		2.77/4.62		1.71	2.21/2.39	~1.6	4.75
**Trp**	4.26			1.9	1.59	2.07/1.56		6.47
**Tyr**	1.84		3.06/5.38		1.99			2.90
**Val**	4.53	0.56	4.34/8.42	2.6	2.64	2.82/6.64	~1.5	5.92

Note. *—calculated according to the authors’ data as the cancer amino acid concentration divided by the corresponding healthy control concentration; **—for well-differentiated/for moderately differentiated cancer; ***—for stage I-II/for stage III-IV; ****—determined from Figure 2 in Ref. [36].

**Table 4 metabolites-13-00950-t004:** Diagnostic characteristics for individual amino acids.

N	Type of Cancer	AAs	AUC	Sensitivity, %	Specificity, %	Cutoff Point (ng/mL)
**1**	OSCC [25]	Val	0.81 (0.706–0.911)	82.4	75.7	-
		Phe	0.64 (0.508–0.765)	52.9	56.8	-
**2**	OSCC [27]	Phe	0.695/0.767 *	84.6/47.1	61.7/95.0	-
		Leu	0.863/0.852 *	84.6/82.4	81.7/80.0	-
**3**	OSCC [33]	Met	0.925	-	-	-
		Leu	0.923	-	-	-
**4**	Breast cancer [34]	Phe	0.748/0.739 *	64.7/70.0	82.1/82.1	599.3/570.3
		Trp	0.763/0.786	82.4/90.0	71.4/71.4	46.1/45.1
		Met	0.786/0.786	82.4/90.0	71.4/71.4	6.8/5.9
		Pro	0.866/0.857	70.6/80.0	92.8/92.9	11,119.7/10,959.1
		Thr	0.830/0.886	76.5/90.0	85.7/85.7	408.3/412.3
		Asp	0.792/0.696	82.4/80.0	67.9/67.9	362.1/360.2
		Ser	0.750/0.832	76.5/90.0	67.9/71.4	931.3/1010.5
		His	0.695/0.646	52.9/50.0	82.1/82.1	1317.9/1317.0
		Gln	0.769/0.832	58.8/90.0	82.1/64.3	852.6/531.6
		Leu	0.748/0.857	76.5/100.0	75.0/71.4	1011.6/959.9
		Val	0.727/0.961	70.6/90.0	71.4/92.8	280.1/532.6
		Glu	0.798/0.861	58.8/90.0	89.3/89.3	1977.4/1925.8
		Lys	0.706/0.821	76.5/80.0	60.7/51.4	3210.8/3807.8
**5**	Breast cancer [35]	Phe	0.739 (0.597–0.881)	77.8	66.7	-
		His	0.847 (0.736–0.958)	96.3	62.5	-
**6**	Thyroid cancer [44]	Gly	0.743 (0.650–0.837)	100.0	51.0	879.6
		Ala	0.814 (0.736–0.891)	72.1	76.5	388.2
		Pro	0.754 (0.665–0.843)	50.8	92.2	1241.7
		Val	0.833 (0.758–0.907)	80.3	78.4	2806.7
		Thr	0.755 (0.663–0.848)	63.9	92.2	198.3
		Leu	0.746 (0.657–0.835)	62.3	76.5	332.5
		Ile	0.689 (0.589–0.789)	86.9	47.1	96.6
		Met	0.678 (0.576–0.779)	90.2	45.1	36.3
		Phe	0.749 (0.658–0.839)	98.4	43.1	592.0
		Trp	0.732 (0.641–0.824)	63.9	76.5	53.7
**7**	Lung cancer [41]	Ile	0.620	-	-	-
		Leu	0.621	-	-	-
		Lys	0.620	-	-	-
		Phe	0.634	-	-	-
		Tyr	0.618	-	-	-
		Trp	0.663	-	-	-
**Mean** **value**			0.748 ± 0.026	75.1 ± 4.9	71.8 ± 5.1	-

Note. *—for stage I–II/for stage III–IV.

**Table 5 metabolites-13-00950-t005:** Methods of multidimensional data processing and diagnostic characteristics of algorithms based on them.

N	Statistical Methods	Type of Cancer	Variables in the Model	Characteristics
**1**	Principal component analysis (PCA); Multiple logistic regression (MLR)	OSCC [24]	**Alanine**, Choline, **Leucine** + **Isoleucine**, **Glutamic acid**, 120.0801 *m*/*z*, **Phenylalanine**, alpha-Aminobutyric acid, **Serine**	AUC—0.865 (0.810)
**2**	Principal component analysis (PCA); Orthogonal partial least squares-discriminant analysis (OPLS-DA); Logistic regression (LR)	OSCC [25]	OSCC vs. HC: Lactic acid and **Valine**	AUC—0.890 (0.813–0.972) Sensitivity—86.5% Specificity—82.4%
			OSCC vs. leukoplakia: Lactic acid, **Phenylalanine**, **Valine**	AUC—0.970 (0.932–1.000) Sensitivity—94.6% Specificity—84.4%
**3**	Logistic regression (LR)	OSCC [27,28,29]	HC vs. T1–2: **Phenylalanine**, **Leucine**	AUC—0.871 (0.767–0.974) Sensitivity—92.3% Specificity—81.7%
			HC vs. T3–4: **Phenylalanine**, **Leucine**	AUC—0.899 (0.827–0.971) Sensitivity—94.1% Specificity—75.0%
**4**	Partial least squares regression-discriminant analysis (PLS-DA)	OSCC [32]	**Histidine**, **Tyrosine**	AUC—0.94 (0.79−1.0)
**5**	Principal component analysis (PCA); Multiple logistic regression (MLR)	Breast cancer [24]	173.0285 *m*/*z*, **Lysine**, 409.2312 *m*/*z*, **Threonine**, **Leucine** + **Isoleucine**, Putrescine, 131.1174 *m*/*z*, **Glutamic acid**, **Tyrosine**, Piperideine, **Valine**, **Glycine**, 437.7442 *m*/*z*	AUC—0.973 (0.881)
**6**	Multiple logistic regression (MLR)	Breast cancer [34]	SFAA index: **Proline**, **Threonine**, **Histidine**	HC vs. T1–2: AUC—0.916 (0.834–0.998) Sensitivity—88.2% Specificity—85.7%
**7**	Multiple logistic regression (MLR); Alternative decision tree (ADTree + Bagging)	Breast cancer [36]	Spermine, ribulose 5-phosphate, 1,3-Diaminopropane, Butanoate, **Threonine**, DHAP, **Leucine**, Cadaverine, GABA, Propionate, N-acetylneuraminate, N1-acetylspermine, **Arginine**, Carnitine, N1-acetylspermidine, Lactate, Ile, Spermidine, **Serine**, Succinate, **Alanine**, gamma-Butyrobetaine, 5-Aminovalerate, Choline, **Glutamine**, **Valine**	AUC—0.912 (0.838–0.961)
**8**	Principal component analysis (PCA); Cluster analysis; Orthogonal partial least squares-discriminant analysis (OPLS-DA); ANN model	Lung cancer [40]	Gamma aminobutyric acid (GABA), Cytosine, Uracil, Creatinine, Pyroglutamic acid, Ketoleucine, Adenine, Imidazolepropionic acid, Allysine, Guanine, 3-hydroxyanthranilic acid, gentisic acid, N-acetylproline, N-acetylhistidine, **Serine**, **Proline**, **Valine**, Phenylglyoxylic acid, Xanthine, **Arginine**, N-acetyl-L-glutamic acid, N-acetyltaurine, Glycylphenylalanine	AUC—0.986 Sensitivity—97.2% Specificity—92.0%
**9**	Multiple logistic regression (MLR)	Lung cancer [41]	Diethanolamine, Cytosine, **Lysine**, **Tyrosine**	AUC—0.663 (0.516–0.810)
	Principal component analysis (PCA); Logistic regression (LR)	Gastric cancer [38]	Taurine, **Glycine**, **Glutamine**, Ethanolamine, **Histidine**, **Alanine**, **Glutamic acid**, Hydroxylysine, **Proline**, **Tyrosine**	Sensitivity > 80% Specificity > 87.7%
**1** **0**	Principal component analysis (PCA); Partial least-squares discriminant analysis (PLS-DA); Binary logistic regression (BLR)	Thyroid cancer [44]	**Alanine**, **Proline**, **Phenylalanine**, **Valine**	AUC—0.936 (0.894–0.977) Sensitivity—91.2% Specificity—85.2%
**11**	Principal component analysis (PCA); Multiple logistic regression (MLR)	Pancreatic cancer [24]	**Phenylalanine**, **Tryptophan**, Ethanolamine, Carnitine, 173.0919 *m*/*z*	AUC—0.993 (0.944)
**1** **2**	Principal component analysis (PCA); Random Forest model with a leave-one-out cross-validation (LOOCV); classification and regression tree method (CART)	Hepatocellular cancer [45]	125 metabolites (RF125)	AUC—0.845 Sensitivity—81.8% Specificity—87.2%
			12 metabolites (iRF12): Octadecanol, Acetophenone, Lauric acid, 1-monopalmitin, Dodecanol, Salicylaldehyde, Glycyl-proline, 1-monostearin, Creatinine, **Glutamine**, **Serine**, 4-hydroxybutyric acid	AUC—0.886 Sensitivity—84.8% Specificity—92.4%
			4 metabolites (iRF4): Octadecanol, Acetophenone, 1-monopalmitin, 1-monostearin	AUC—0.917 Sensitivity—87.9% Specificity—95.5%
			CART: Octadecanol, Acetophenone, 1-monopalmitin, 1-monostearin	AUC—0.907 Sensitivity—87.9% Specificity—93.5%
**1** **3**	Partial least squares-discriminant analysis (PLS-DA); Multiple logistic regression (MLR); ADTree algorithm	Colorectal cancer [46]	N-acetylputrescine, N1-acetylspermine, N1,N8 -diacetylspermidine, N8-acetylspermidine, N1 -acetylspermidine, N1,N12-diacetylspermine, pyruvate, lactate, succinate, malate, 4-methyl-2-oxopentanoate, 5-oxoproline, **Isoleucine**, **Valine**, **Lysine**, **Alanine**, 3-Aminoisobutyrate, alpha-Aminoadipate, 2AB, Cadaverine, 2-Hydroxy-4-methylpentanoate, gamma-Butyrobetaine, Creatine	CRC + AD vs. HC AUC—0.870 (0.837–0.903)

Note. The amino acids included in the diagnostic algorithms are highlighted in bold.

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
