# Peer review of "Diagnostic Value of Salivary Amino Acid Levels in Cancer"

_metabolites, 2023, doi:10.3390/metabo13080950_

Round 1
Reviewer 1 Report
In the manuscript "Amino acid profile of saliva in oncological diseases" by Bel’skaya et al., the authors describe a data obtained from the papers about amino acids in saliva and cancer.
The major.
1) The title does not fit the content of the manuscript. The authors themselves claim that
P19… “a logical and complete substantiation of the changes in saliva occurring in cancer, including changes in the amino acid profile, has not yet been formed” …
P52… “The existing literature data on the content of amino acids in saliva in cancer patients are few, scattered and contradictory”.
Actually, the authors present profiles of metabolites (Table 4) including amino acids. So it looks like amino acid analysis can be statistically relevant only in combination with different metabolites, as the authors say themselves.
P139… “…when using combinations of amino acids with each other or with other metabolites, sensitivity and specificity values increase (Table 4 , 5).
2) To talk about amino acid profiles, absolute concentrations of amino acids in saliva are necessary.
The minor
There are a lot of abbreviations in the text. Abbreviation list should be included.
English should be brushed
Reviewer 2 Report
Finding new tumor biomarkers from liquid biopsies and their analysis is an important topic for early cancer detection. This manuscript has analyzed the composition of amino acids in the saliva of different types of tumors. When I was reading this manuscript, it is not understandable, why it is written that way at all. It is clear, that in the literature is more data about amino acids in the saliva for the oral squamous cell carcinoma. The selection of tumors is random and there is no general concept. Ma be better is to choose cancers from the same histological type, body tissue type or for example only for one organ system like digestive system.
Abbreviations should be given separately.
A lot of tables are used in the work. For example, table 3 could be made into a figure.
Associations between cancer stage and amino acid profile have only been reported in breast cancer. Are such data also available for any other tumors?
Which amino acids would be suitable candidates for the early diagnosis of cancer in the described tumor types?
How the DAA is calculated?
Reviewer 3 Report
The Authors deliberately omitted the topic of saliva sampling, which was an incorrect assumption (lines 120-121, 238). By not including this information and by not referring to mutually exclusive results, they fail to show the full complexity of the topic. In the case of saliva, the timing of its collection and, above all, the manner in which this was done is crucial in its composition both qualitatively and quantitatively. Therefore, the authors must address this very issue. And the paper must be complemented by the methodology of saliva collection, especially in the case of contradictory results. And this information must appear in the discussion.
Contrary to the information provided (line 56), there are no specific standards for saliva sampling, which is why it is important to indicate which sampling method was used. And the indicated publications describe specific amino acids and certainly cannot be regarded as guidelines.
Results
The presentation of the methods used for the determination of amino acids is very chaotic. Whether HPLC, UPLC or RPLC, it is still liquid chromatography and it is the type of chromatography (GC or LC) that should determine the division of methods. It makes no sense to list the types of mass spectrometers, these will still be MS. The details could be included in the table, but mixing this in the text makes it unreadable and creates unnecessary chaos.
Table 2 - '+' signs appear in the table itself - what is the purpose of their use? If they denote the determination itself, they are unnecessary, because next to them is information about the increase or decrease in concentration of the amino acid, so it is known that it was determined anyway.
Table 2 should be at the end of the manuscript as a summary of the information on the change in concentration of the amino acids concerned, but should be supplemented with the type of tumour in which these changes were found. Or a similar summary table should be included at the end.
Reviewer 4 Report
The manuscript reviews several other research papers that examined amino acid levels in saliva from samples from patients with various cancers.
It is a good summary of the results, but could be improved in its presentation style.
1) In the introduction it would help for the authors to describe how amino acid levels in saliva interact with their levels in cells. Make the connection to how the amino acid levels in a tumor can correlate with those in saliva.
2) Table 2 would be much better to read if the cancers associated with each column were listed on the table rather than having to go to the references.
3) In the discussion, a couple of figures showing amino acid metabolism would help the reader as the read the paragraph on amino acid metabolism.
The English is good. There area a couple of errors, but overall it reads quite well.
Round 2
Reviewer 1 Report
Still, one major point should be explained.
1) The issue about absolute concentrations of amino acids in saliva still needs to be discussed in more details. What is the solution for this diagnostic problem, keeping in mind the wide range of amino acid concentrations? There is incorrect citation in the text - “…for example, in [30], the concentration of alanine was 11424.5±12290.0 ng/ml, leucine - 1072.3±1344.7 ng/ml, etc.” Actually should be [34] not [30].
The minor
2) There is no need to include the definition of the standard abbreviations for amino acids.
The usage of commas should be checked all around the text.
Reviewer 3 Report
The Authors responded to all comments. The work is definitely more readable.